Applying fecal microbiota transplantation (FMT) to treat recurrent Clostridium difficile infections (rCDI) in children

Fareed Shaaz 1 4
Sarode Neha nehasarode@fas.harvard.edu 2
Stewart Frank J. 3
Malik Aneeq 4
Laghaie Elham 4
Khizer Saadia 5
Yan Fengxia 6
Pratte Zoe 3
Lewis Jeffery 7
Immergluck Lilly Cheng limmergluck@msm.edu 1 4
1 Department of Microbiology/Biochemistry/Immunology, Morehouse School of Medicine , Atlanta , GA , United States of America
2 Department of Organismic & Evolutionary Biology, Harvard University , Boston , MA , United States of America
3 School of Biological Sciences, Georgia Institute of Technology , Atlanta , GA , United States of America
4 Clinical Research Center, Morehouse School of Medicine , Atlanta , GA , United States of America
5 Clinical Research, Children’s Healthcare of Atlanta , Atlanta , GA , United States of America
6 Department of Community Health & Preventive Medicine, Morehouse School of Medicine , Atlanta , GA , United States of America
7 Pediatric Gastroenterology, Children’s Center for Digestive Health Care, LLC , Atlanta , GA , United States of America
Crandall Keith
Electronic publication date: 2018 May 30
Publication date: 2018
Volume: 6
Electronic Location ID: e4663
Received 2017 Dec 10; Accepted 2018 Apr 2
Copyright: ©2018 Fareed et al.
Copyright year: 2018
Copyright holder: Fareed et al.
License: This is an open access article distributed under the terms of the Creative Commons Attribution License, which permits unrestricted use, distribution, reproduction and adaptation in any medium and for any purpose provided that it is properly attributed. For attribution, the original author(s), title, publication source (PeerJ) and either DOI or URL of the article must be cited.
License URL: https://creativecommons.org/licenses/by/4.0/

Keywords: Fecal microbiota transplantation, Pediatrics, Clostridium difficile infection, Microbiome

Funding: Children’s Healthcare of Atlanta, Friends’ Fund National Center for Advancing Translational Sciences of the National Institutes of Health UL1TR002378 K-08 Agency Healthcare Research Quality HS024338-01 Centers for Disease Control and Prevention Emerging Infections Program Grants U01C10000307-05 U01000312 This work was supported by the Children’s Healthcare of Atlanta, Friends’ Fund; National Center for Advancing Translational Sciences of the National Institutes of Health under Award Number UL1TR002378; K-08 Agency Healthcare Research Quality HS024338-01; Centers for Disease Control and Prevention Emerging Infections Program Grants U01C10000307-05 and U01000312. The funders had no role in study design, data collection and analysis, decision to publish, or preparation of the manuscript.

==============================
Background

Fecal Microbiota Transplantation (FMT) is an innovative means of treating recurrent Clostridium difficile infection (rCDI), through restoration of gut floral balance. However, there is a lack of data concerning the efficacy of FMT and its impact on the gut microbiome among pediatric patients. This study analyzes clinical outcomes and microbial community composition among 15 pediatric patients treated for rCDI via FMT.

Methods

This is a prospective, observational, pilot study of 15 children ≤18 years, who presented for rCDI and who met inclusion criteria for FMT at a pediatric hospital and pediatric gastroenterology clinic. Past medical history and demographics were recorded at enrollment and subsequent follow-up. Specimens of the donors’ and the patients’ pre-FMT and post-FMT fecal specimen were collected and used to assess microbiome composition via 16S rRNA gene sequencing.

Results

FMT successfully prevented rCDI episodes for minimum of 3 months post-FMT in all patients, with no major adverse effects. Three patients reported continued GI bleeding; however, all three also had underlying Inflammatory Bowel Disease (IBD). Our analyses confirm a significant difference between pre-and post-FMT gut microbiome profiles (Shannon diversity index), whereas no significant difference was observed between post-FMT and donor microbiome profiles. At the phyla level, post-FMT profiles showed significantly increased levels of Bacteroidetes and significantly decreased levels of Proteobacteria. Subjects with underlying IBD showed no difference in their pre-and post-FMT profiles.

Conclusion

The low rate of recurrence or re-infection by C. difficile, coupled with minimal adverse effects post-FMT, suggests that FMT is a viable therapeutic means to treat pediatric rCDI. Post-FMT microbiomes are different from pre-FMT microbiomes, and similar to those of healthy donors, suggesting successful establishment of a healthier microbiome.

Introduction

Childhood infections by the anaerobic gram-positive bacterium Clostridium difficile pose a significant health challenge, with limited viable treatment options for recurrent infections. Approximately half a million Clostridium difficile infections (CDI) occur annually in the United States and are associated with approximately 29,000 deaths (Lessa et al., 2015). Although an estimated 17,000 CDI cases involve children ages 1 to 17 years (Center for Disease Control and Prevention, 2015), the majority of studies exploring treatment efficacy focus on adults (Lees et al., 2016). Even though clinical disease is not seen frequently among infants and very young children, presumably due in part to the lack of toxin-binding receptors in this age group coupled with protective maternal antibodies, the incidence among children has continued to increase over the past two decades (Nylund et al., 2011; Zilberberg, Tillotson & McDonald, 2010). Risks associated with CDI include prior antibiotic use (Schutze et al., 2013) and gut motility dysfunction, with higher incidence observed in children with inflammatory bowel disease (IBD) (Pascarella et al., 2009). Recurrence of disease also has increased, with rates estimated to be as high as 25% (Aslam, Hamill & Musher, 2005). Currently, the preferred first line treatment for recurrent CDI (rCDI) is oral vancomycin (Leong & Zelenitsky, 2013; Sandora et al., 2011). However, up to 25% of patients undergoing oral vancomycin therapy for rCDI relapse within one month of treatment (Louie et al., 2011; Vincent & Manges, 2015). Furthermore, dysbiosis in gut commensal bacteria, caused initially by C. difficile infection, is further exacerbated by antibiotic use (Isaac et al., 2017).

The human gut microbiome is a topic of extensive interest given overwhelming evidence for the microbiome’s role in shaping health and disease (Dore & Blottiere, 2015; Estrela, Whiteley & Brown, 2015; Lozupone et al., 2012). Bacteroidetes and Firmicutes are the major gut phyla of interest: they compose >90% of the healthy human gut microbiome, and as commensal bacteria, function as a barrier in preventing the C. difficile overgrowth (Bien et al., 2013; Malys, Campbell & Malys, 2015). Studies profiling rCDI gut microbiomes show overall decreased species richness, decreased ratios of Bacteroidetes to Firmicutes, and increased Proteobacteria, (Chang et al., 2008; Theriot & Young, 2015) suggesting attempts to correct microbiome dysbiosis as a potential treatment strategy for rCDI.

For children, the repeated bouts of diarrhea, abdominal pain, and medical treatment associated with rCDI incur costs and interfere with normal function. One innovative approach to re-establish the gut microbiome is through fecal microbiota transplantation (FMT). In a review of 317 adult rCDI patients (average age of 53 years) treated via FMT, FMT alleviated rCDI symptoms in 92% of cases (Gough, Shaikh & Manges, 2011). Furthermore, FMT has been shown to successfully address dysbiosis by improving the ratio of Bacteroidetes and Firmicutes, thus producing a post-FMT gut bacterial community similar to that of the donor (Seekatz et al., 2014). In general, the pediatric microbiomes not only differ in the proportions of major gut phyla (Hollister et al., 2015), but also exhibit significantly higher interpersonal variation and lower diversity compared to adult microbiomes (Yatsunenko et al., 2012). Though FMT has showed promising results in adults (Rao & Safdar, 2016), this treatment option has been applied sparingly in children (Russell et al., 2010), and even fewer studies have analyzed the efficacy of FMT to treat pediatric rCDI and the impact of the procedure on pediatric gut microbiomes (Hourigan & Oliva-Hemker, 2016).

The primary goal of this prospective, observational and epidemiological pilot study was to assess the practicality, safety and efficacy of FMT as a treatment option for pediatric patients with rCDI. A secondary aim was to observe the gut microbiota changes elicited by FMT within pediatric rCDI patients. There have been 45 cases using FMT in children with rCDI and the largest series involved 10 children, three of whom had IBD (Kronman et al., 2015). To our knowledge, ours is the largest FMT study to be conducted among pediatric patients with rCDI, which involves children both with and without underlying inflammatory bowel disease.

Materials & Methods

Overall study design

This is an observational epidemiology study to look at children who failed multiple courses of standard antibiotic therapy for rCDI and elected to undergo FMT. Prior to sampling, this study was approved by the institutional review board of Children’s Healthcare of Atlanta and Morehouse School of Medicine.

Study participants (n = 15 pediatric patients) were screened and recruited from a pediatric hospital (Children’s Healthcare of Atlanta Hospital at Scottish Rite) and a private pediatric gastroenterology clinic (Children’s Center for Digestive Health Care, LLC) in Atlanta, Georgia. All study participants were children (age range 21 months to 18 years) who presented with symptoms consistent with rCDI, as determined from examination by the same pediatric gastroenterologist. Patients were eligible for the FMT procedure if they fit the study definition of rCDI: ongoing diarrhea, a positive C. difficile fecal specimen test (polymerase chain reaction assay to detect C. difficile toxin genes), and history of failed CDI treatment with antibiotics for a minimum of two prior CDI episodes, of which, at least one episode was treated with a course of oral vancomycin within the previous eight weeks (Cohen et al., 2010). All FMT procedures were performed via either colonoscopy or nasojejunal tube by the same pediatric gastroenterologist. Previous episodes of CDI were recorded if patient had a documented positive C. difficile toxin PCR test and a diagnosis of CDI in his/her medical records. Informed consent (or assent when age appropriate) was obtained from a patient’s parent or legal guardian prior to the FMT procedure. A survey questionnaire was administered in a personal interview from study staff to record participant demographic information (age, gender, race/ethnicity), underlying chronic health conditions, prior antibiotic use within 12 months of the time of enrollment, and CDI related symptoms. The following information was also collected from each participant’s medical records: nature of prior episodes of CDI, dates of positive PCR C. difficile toxin tests, and provider notes documenting history of infections that required antibiotics and specific symptoms consistent with CDI. Five donor samples were ordered from OpenBiome™ (Medford, MA, USA) to be used for the FMT procedure. All donors were non-family related and underwent laboratory screening by Openbiome™ (Osman et al., 2016; Osman et al., 2016) for infectious risk factors (including Multi-drug resistant organisms (MDRO’s), serologic testing, and fecal specimen testing) and potential microbiome-mediated conditions.

FMT procedure

FMT procedure was performed on subjects via colonoscopy (n = 14) and nasojejunal tube (n = 1). Prior to FMT, all patients undergoing a colonoscopy completed a standard bowel prep using Polyethylene Glycol 3350. Vancomycin was discontinued at least 48 h prior to the FMT. Colonoscopies were performed under general anesthesia or under conscious sedation. Up to 240 ml of donor fecal specimen was instilled into the terminal ileum and cecum during colonoscopy in 14 of 15 patients. In one patient, who had undergone colonoscopy due to an underlying disease, 60 cc of donor fecal specimen was instilled via nasojejunal tube into the proximal jejunum.

Follow-up

Follow-up telephone calls were made by research staff after 24 h and one-week post-FMT procedure. Survey questionnaires were administered by research staff at one month and ≥3 months post-FMT time points, alongside scheduled evaluation by single pediatric gastroenterologist. Responses to questions about study participant’s symptoms, including fever, vomiting, abdominal pain and distention, evidence of allergic reaction or bloody fecal specimens were recorded in a relational database. If patients were not available to respond at the three-month follow-up visit, attempts to collect a response were performed for up to 15 months after the procedure. (Although fecal samples were collected outside of the three-month study period, patients were not clinically re-evaluated for evidence of rCDI beyond three months of the date of fecal microbiota transplantation.)

Fecal specimen collection

Three fecal specimen samples were collected for each study participant: (1) from the donor—used for transplant, (2) from the study participant before transplant (pre-FMT), (3) from the study participant following the FMT procedure (at least 3 months after FMT procedure), with the latter used to analyze the effect of FMT on gut microbiome composition. Fecal samples were de-identified and stored in a −80 °C freezer until DNA extraction.

Fecal specimen DNA extraction

DNA was extracted from fecal specimen samples using the MO BIO PowerSoil DNA extraction kit (MOBIO, Carlsbad, CA, USA) following manufacturer instructions, with modifications as recommended by the Human Microbiome Project (HMP) Manual of Procedures (version 12.0). Briefly, before step 2 of the standard protocol, the PowerBead tubes containing samples were heated at 65 °C for 10 min, followed by a second incubation at 95 °C for 10 min to improve cell lysis. The manufacturer protocol was followed for the remainder of the procedure, except for step 12, during which the tubes were centrifuged for 2 min. DNA was extracted from triplicate subsamples for each fecal specimen sample, with the extracted DNA then pooled for quantification and processing. DNA concentrations were measured using the Qubit 2.0 DNA quantification system (ThermoFisher Scientific, Waltham, MA, USA).

16S rRNA gene PCR and amplicon sequencing

Microbiome taxonomic composition in donor, pre-FMT, and post-FMT fecal specimen was characterized by Illumina sequencing of PCR amplicons encompassing the V4 hypervariable region of the 16S rRNA gene. Amplicons were synthesized using Platinum PCR Supermix (ThermoFisher Scientific, Waltham, MA, USA) with V4-specific primers F515 and R806 (Caporaso et al., 2010), and uniform amounts of DNA template per reaction. Both forward and reverse primers were barcoded and appended with Illumina-specific adapters, as per instructions by Kozich et al. (2013). Thermal cycler conditions were as follows: initial denaturation at 94 °C (3 min), followed by 30 cycles of denaturation at 94 °C (45 s), primer annealing at 55 °C (45 s) and primer extension at 72 °C (90 s), and a final extension at 72 °C for 10 min. Amplicons were verified on agarose gel electrophoresis for size (∼400 bp) and purified using Diffinity RapidTip2 pipet tips (Diffinity Genomics, West Chester, PA, USA). Barcoded and Illumina adaptor-appended amplicons for each sample were pooled at equimolar concentrations and sequenced on an Illumina MiSeq using a 500 cycle v2 reagent kit (250 bp paired end) with 15% PhiX genomic library addition to increase read diversity.

Statistical and diversity analyses

Demultiplexed amplicon read pairs were quality trimmed with Trim Galore! (Babraham Bioinformatics, Cambridge, UK), using a base Phred33 score threshold of Q25 and a minimum length cutoff of 100 bp. High quality paired reads were then merged using the software FLASH (Magoč & Salzberg, 2011). Merged reads were analyzed using QIIME v1.9.0 (Caporaso et al., 2010), according to standard protocols. Briefly, reads were screened for chimeras using QIIME’s identify_chimeric_seqs.py script with Usearch61 (Edgar, 2010). Non-chimeric sequences were clustered into Operational Taxonomic Units (OTUs) at 97% sequence similarity using UCLUST (Edgar, 2010) based on open-reference OTU picking with the script pick_open_reference_otus.py. Taxonomy was assigned to representative OTUs from each cluster using the Greengenes database (Aug 2013 release) (DeSantis et al., 2006).

R (R Core Team, 2017) packages Phyloseq (McMurdie & Holmes, 2013) and Vegan (Jari Oksanen et al., 2017) were used to measure alpha and beta diversity, respectively. Alpha diversity was estimated by the number of observed unique OTUs and the Shannon diversity index at an even sampling depth of 27,700 sequences. OTUs observed across at least 70% samples were used for identifying shared OTUs between groups. Microbiome compositional differences among samples were assessed using non-metric multidimensional scaling (NMDS) based on Bray-Curtis distances at OTU level.

The significance of the effect of group status (donor, pre-FMT, post-FMT) on microbiome composition was assessed using the envfit function in Vegan, with 999 permutations. Wilcoxon signed rank test with Benjamini–Hochberg correction for multiple hypothesis testing was used to compare the taxonomic differences between pre- and post- FMT, pre- FMT and donor as well as post-FMT and donor. The differences between IBD and non-IBD were compared using Wilcoxon two sample test for demographic information (age, gender, race/ethnicity), underlying chronic health conditions, prior antibiotic use within 12 months of the time of enrollment, and CDI related symptoms and participant’s medical records: Nature of prior episodes of CDI, dates of positive PCR, C. difficile toxins, and provider notes documenting history of infections which required antibiotics and specific symptoms consistent with CDI. R and SAS 9.4 were used to perform all the statistical tests and adjusted p-value < 0.05 (after multiple hypothesis correction) was reported as statistically significant.

Results

Patient enrollment/population characteristics

Fifteen pediatric study participants (eight female, seven male) were enrolled in this study. Of the potentially 45 specimens, only 40 fecal bio specimens were collected from these 15 patients. (Three post-FMT fecal samples were lost to follow up (LTFU), and two Pre-FMT samples did not yield enough DNA for analysis.) The median age of patients was eight years (range: 21 months to 18 years) (Table 1). Patients had an average of three episodes of CDI prior to FMT (range: two to five CDI episodes.). On average, post-FMT fecal samples were collected 6 months after the FMT procedure (range: three to 14 months) (Fig. 1).

Table 1 Patient characteristics.

Summary of patient profiles for all subjects enrolled in this study. Included are demographic information, delivery route of fecal microbiota transplantation (FMT), underlying inflammatory bowel disease (IBD), and recorded number of prior Clostridium difficile infections (CDI) episodes.

ID	Sex	Race	FMT delivery method	Age (years) at FMT	Time FMT and post-FMT stool collection (Months)	Number of antibiotic courses prior to FMTc	Underlying IBD	Number of CDI episodes prior to FMT	FMT donor ID	
1	F	Biracial	Colonoscopy	2	3	4	No	4	05	
2	F	Black	Colonoscopy	7	9	1	Ulcerative Colitis	3	05	
3	M	White	Colonoscopy	16	4	4	Ulcerative Colitis	4	05	
4	F	Biracial	Colonoscopy	8	14	3	No	3	05	
5	M	Black	Colonoscopy	10	7	4	Ulcerative Colitis	3	05	
6	F	Black	Nasojejunal Tube	8	LTFUa	8	Ulcerative Colitis	3	05	
7	M	White	Colonoscopy	1	6	4	No	4	37	
8	F	White	Colonoscopy	7	4	2	Crohn’s Disease	3	37	
9	F	White	Colonoscopy	18	4	2	No	2	37	
10	F	White	Colonoscopy	2	6	2	No	3	37	
11	F	White	Colonoscopy	8	6	2	No	5	37	
12	M	White	Colonoscopy	2	LTFU	2	No	2	12	
13	M	Biracialb	Colonoscopy	5	3	3	No	3	77	
14	F	White	Colonoscopy	15	LTFU	7	No	3	66	
15	M	White	Colonoscopy	10	2	1	No	3	77	
Notes.

a LTFU = Sample Lost to Follow Up.

b Biracial = Participant self-identified as Black/White.

c Number of antibiotic courses only within the period of 12 months prior to FMT was included.

All patients had up to date vaccinations for routine immunizations, including Diphtheria-Tetanus-acellular Pertussis, Haemophilus influenzae Type B, Influenza, Pneumococcal Conjugate Vaccine, with seven or 13 Serogroups, Inactivated Polio Virus, Hepatitis A and B; Rotavirus; and Varicella.

Figure 1 Time of stool collection post-fecal microbiota transplantation (FMT).

Displayed is a histogram detailing how many patients submitted stool at the displayed time periods (in months) Post-FMT.

Clinical outcomes

All 15 patients tolerated the FMT procedure without complications immediately post procedure. For the 12 patients who completed their three-month follow-up appointments, all experienced clinical resolution of CDI and had no recurrent episodes within the duration of the follow-up period (three months). Abdominal pain was reported by five patients (none had underlying IBD) at the three-month visit; however, the treating pediatric gastroenterologist attributed the pain to other causes, including functional pain and irritable bowel syndrome not related to CDI. All six patients with underlying ulcerative colitis (n = 5) or Crohn’s disease (n = 1) reported hematochezia and ongoing abdominal distension beyond three months of their FMT date.

Microbiome analysis—overall bacterial community diversity and richness

FMT significantly altered gut microbiome composition in all the participants. Alpha diversity (within sample diversity), measured as both observed OTU richness and Shannon diversity, differed between all compared FMT groups, with diversity in the donor sample higher compared to the matched pre- and post-FMT samples collected from recipients; pre-FMT fecal samples had the lowest diversity. There was no significant difference in observed OTU richness between the five donor samples (Observed OTU p-value = 0.07894; Shannon diversity p-value 0.04306). Post-FMT microbiomes, although more diverse compared to pre-FMT microbiomes, were less diverse than donor microbiomes (Fig. 2A). A significant difference (Kruskal-Wallis test, p-value < 0.001) was also observed in shared OTU’s between all three groups (donor, recipient pre-FMT, and recipient post-FMT), where within each comparison, donor vs. post-FMT profiles showed the smallest variance (calculated using Wilcoxon signed rank test), while pre- vs post-FMT profiles had the greatest variance.

Figure 2 Alpha and beta diversity of samples.

(A) Alpha diversity is a measure of species richness within a sample which is quantitatively expressed here as a box-plot of the number of observed unique taxa/OTU’s and Shannon diversity index on y-axis. Individual samples are colored according to their patient ID grouped by their FMT status on the x-axis. (B) Beta diversity is a measure of taxonomic composition diversity between sample that is represented as Non-metric multidimensional scaling (NMDS) ordination of samples based on Bray-Curtis distance matrix. The color and shape of samples are according to their FMT status identity (RED, donor; BLUE, pre-FMT or GREEN, post-FMT), with samples belonging to the same FMT group connected to form polygons. The more similar the groups are to each other (donor and post-FMT), the closer their polygon clusters are going to be on the ordination plot and vice versa (pre-FMT compared to both donor and post-FMT). The dotted lines connect individual samples to the group centroid while the ellipse gives an estimate of standard deviation of the scores. (FMT, Fecal Microbiota Transplantation).

FMT status (donor, recipient pre-FMT, and recipient post-FMT) also significantly influenced microbiome beta diversity (between sample diversity), as measured by Bray-Curtis distance-based NMDS ordinations (r2 = 0.65, p-value < 0.001; envfit function in Vegan), with post-FMT microbiome composition being more like that of donor than that of the pre-FMT group (Fig. 2B). Together, these results indicate a positive shift in microbial community composition in response to FMT, suggesting a restructuring to a non-disease state akin to the donor.

Microbiome analysis—taxon level changes

The relative representation of major bacterial phyla of interest varied substantially in response to FMT (Fig. 3). Notably, the average ratio of Bacteroidetes to Firmicutes (B:F), the predominant phylum-level indicator of a ‘normal/healthy’ gut microbiome, increased over 7-fold from pre-FMT (average B:F of 0.15) to post-FMT microbiomes (average B:F: 1.12), with the recipient post-FMT B:F approaching that of donor communities (average B:F: 1.49) (Table 2). The shift in this ratio was driven largely by changes in the proportional representation of Bacteroidetes, which comprised an average of only 7% relative abundance across all pre-FMT samples (excluding study participant 15, who was an outlier), but showed a relative abundance of 41% and 57% in recipient post-FMT and donor samples, respectively (Table 2). Much of the observed variation within the Bacteroidetes was due to proportional shifts involving a single genus, Bacteroides. In addition to Bacteroidetes, the phyla Proteobacteria and Verrucomicrobia were significantly enriched in donor and post-FMT compared to pre-FMT fecal samples (Fig. 3), whereas these three phyla did not differ in abundance between donor and post-FMT fecal samples, highlighting a shift in the microbiome profile, likely caused by FMT, towards that of the donor.

Figure 3 Distribution of Major Phyla between Donor, Recipient Pre- and Post-FMT Microbiomes.

The bubble plot above displays the proportions of phyla observed in each sample. Red indicates donor samples, blue indicates pre-FMT samples, and green indicates post-FMT samples. An increase in Bacteroidetes can be observed from pre to post-FMT samples, as well as a decrease in Proteobacteria. Post-FMT phyla proportions are more similar to donor profiles. (FMT, Fecal Microbiota Transplantation).

Table 2 Microbiome analyses- taxon level changes between Bacteroidetes, Firmicutes, and Proteobacteria.

Abundance of major microbial phyla in stool microbiomes, expressed for each sample type as the percentage of total sequences classifiable to the phylum level, averaged across all samples.

Phylum	Donor	Pre-FMT	Post-FMT	
Bacteroidetes	57%	7%	41%	
Firmicutes (F)	38%	50%	36%	
Proteobacteria (B)	4%	41%	5%	
Average B:F Ratio	1.49	0.15	1.12	

Microbiome analysis—bacterial profile by genus

The value of examining the proportional representation of microbial phyla with regards to gut microbiome health has been questioned (Jandhyala et al., 2015). When examining the changes that occurred instead at the genus level, the most noticeable shifts were reflective of the changes that occurred at the phylum level (Table 2). Specifically, the genus Bacteroides showed a considerable and significant increase in recipients’ pre- to post-FMT (from average 6.5 to 39.2 %). The average relative abundance of the genus Clostridium in the donor was 0.02% (min = 0.001%, max = 0.08%), consistent with a non-disease state. In response to FMT, the average relative abundance of Clostridium in FMT recipients fell roughly 80%, from an average of 1% pre-FMT (min = 0%, max = 4.9%) to 0.16% post-FMT (min = 0%, max = 1.58%). Three of the FMT recipient participants had Clostridium relative abundance of 0%, four had very low but detectable percentages (≤0.01%), while 6 had levels ≥0.5%. Genera identified to be altered significantly in relative abundance per FMT status are listed in Table S1.

Effect of underlying IBD

Of the 15 patients enrolled, 10 had no underlying inflammatory bowel disease (IBD), whereas five had been diagnosed with IBD (four ulcerative colitis and one Crohn’s disease with one patient having had total colectomy) (Table 3). Community taxonomic profiles clustered according to pre- versus post-FMT status, rather than IBD status (Fig. 4), suggesting that underlying IBD had little to no effect on the microbial shift observed because of FMT. There was a marginal effect in the Proteobacteria abundance on separation of community with and without IBD (r2 = 0.54, p = 0.018).

Table 3 Comparison of symptoms post fecal microbiota transplantation (FMT) for patients with and without underlying inflammatory bowel diseases (IBD).

Gastrointestinal (GI) symptoms include one or more of the following: abdominal pain, fever, abdominal distension, or bloody stool. (The categorical variables were compared using Fisher’s Exact test due to the small sample and the continuous variables were evaluated using Wilcoxon two independent sample test.)

Variable	No underlying IBD (n = 10, 66%)	Underlying IBD (n = 5, 33%)	p-value	
Age, median(IQR)	6.55(8.01)	8.02(2.61)	0.4053	
Gender male%	5(50)	2(40)	1.0000	
GI symptoms prior to FMT (yes, %)	9(90)	5(100)	1.0000	
GI symptoms at 24-hour post FMT			0.5055	
No	3(30)	0(0)		
Yes	7(70)	4(100)		
GI symptoms at 1-week post FMT			0.0667	
No	1(12.5)	3(75)	 	
Yes	7(87.5)	1(25)	 	
GI symptoms at 1-month post FMT			0.2657	
No	3(37.5)	4(80)	 	
Yes	5(62.5)	1(20)	 	
GI symptoms at 3 months post FMT			1.0000	
No	2(28.6)	1(20)	 	
Yes	5(71.4)	4(80)	 	
Route of FMT			0.3333	
Colonoscopy	10(100)	4(80)	 	
NJ Tube	0	1(20)	 	
Number of antibiotic courses prior to FMT, median(IQR)	2.5(2)	4(2)	0.6217	
Number of ED visits within a year of FMT, median(IQR)	1.5(3)	4(3)	0.0468	
Number of CDI episodes, median(IQR)	3(1)	3(0)	0.9456	
B:F Ratio, median(IQR)	0.0021(0.0068)	0.0030(0.0027)	0.8209	
Clostridia Distribution, median (IQR)	0.0055(0.0127)	0.00001(0.0109)	0.4102	

Figure 4 Effect of inflammatory bowel disease (IBD) on community diversity.

NMDS ordination of samples considering their IBD status showed that, underlying IBD condition had no significant effect on community dynamics. Samples are colored based on the combination of their FMT and IBD status, with samples belonging to the same group connected with polygon for visual clarity. (FMT, Fecal Microbiota Transplantation).

Furthermore, no discernable differences in post-procedural symptoms such as abdominal pain, vomiting, or bloody fecal specimens were observed between patients with or without underlying IBD.

Discussion

Fecal microbiota transplantation (FMT) is a novel approach to treating rCDI by restoration of a healthy gut microbiome (Bakken et al., 2011). FMT has become an acceptable therapeutic consideration for adults, particularly after recurrence of CDI (Surawicz et al., 2013). However, few data are available from pediatric studies beyond case reports or small case series for children (Hourigan & Oliva-Hemker, 2016). Our findings are consistent with what has been reported in adults and children who have undergone FMT using non-familial, non-autologous donors. Specifically, in our study of 15 children, we show that this procedure is well tolerated by children, irrespective of underlying gastrointestinal disorders (ulcerative colitis and Crohn’s) and that the taxonomic shift in the gut microbiome to resemble the donor’s microbiome is similar regardless of compositional variation in pre-FMT profiles among patients. Moreover, resolution of CDI symptoms seemed to last from at least three months to as long as 14 months.

Very young children with rCDI represent a challenging population, given that the microbial gut community has not ‘matured’ and the diversity of dominant phyla is shifting. In our study, almost 25% of participants were less than two years of age (one of the four patients had ulcerative colitis and received FMT at 21 months of age). However, we did not identify any significant differences in the distribution of dominant phyla across the different ages. There are limited data on how FMT (usually transplantation is with adult donors, who have ‘mature’ microbial profiles) might affect those as young as 2 years of age (Kahn, Gorawara-Bhat & Rubin, 2012; Kronman et al., 2015; Russell et al., 2014; Walia et al., 2014; Wang et al., 2015).

Similar to what has been observed in larger adult studies (Seekatz et al., 2014) we also found that for most of our pediatric patients, FMT seemed to induce a post FMT microbiome that was significantly different from the pre-FMT microbiome profile, and more similar to the healthy donor profile. It would be interesting to explore serial fecal biospecimens for up to one year post FMT and compare the trends of the microbial community composition, particularly as it relates to the Bacteroidetes: Firmicutes ratio for both patients with and without underlying gastrointestinal disorders.

Overall, pre-FMT profiles showed high abundance of Proteobacteria, with low amounts of Bacteroidetes. Post-FMT profiles showed an increase in Bacteroidetes, with a large decrease in Proteobacteria. These findings are in agreement with results from past studies of adult patients (Seekatz et al., 2014), De’Argenio and Salvatore, for example, found that a high abundance of Proteobacteria pre-FMT is associated with underlying conditions, such as ulcerative colitis and Crohn’s disease (D’Argenio & Salvatore, 2015). Furthermore, Proteobacteria is often selected over Bacteroidetes and Firmicutes when colitis is present (Bradley & Pollard, 2017). It is possible that these factors were responsible for the distributions of Proteobacteria observed in this study.

With regards to major gut phyla, two patients showed a decrease in the relative abundance of Bacteroidetes from pre- to post-FMT. Although this was not expected, as all other profiles showed an often, large increase in Bacteroidetes abundance, these two patients also demonstrated significant decreases in the abundances of Proteobacteria, thus still appearing more similar to their respective ‘healthy’ donor profiles. It is likely that an increase in Bacteroidetes was not observed since the pre-FMT profiles of these patients already displayed higher abundances of Bacteroidetes in comparison to the other subjects.

Interestingly, underlying gastrointestinal conditions did not adversely affect tolerance of the procedure or preponderance of post procedural adverse effects, nor were there any significant differences observed in pre-FMT profiles or distribution of Bacteroidetes, Firmicutes, or Proteobacteria; the shift in major gut phyla post-FMT was similar across all patients.

Limitations/future directions

Because this was a prospective observational pilot study, microbiome results and statistical implications are limited by a low sample size (n = 15 patients). Not all follow-up phone calls were completed within the time frame set in the study design, and 5 post-FMT samples were not obtained. Additional variables that could impact the microbiome, such as diet, exercise, environmental factors, puberty, etc., were not collected, although the list of such potential factors could be in practice rather exhaustive. Fecal samples were ultimately analyzed three to 12 months’ post-procedure due to inconsistencies in patient compliance. Our covariate analysis is also limited by the fact that study surveys were self-reported. We recognize the risks associated with FMT, and although this study shows promising results, we suggest that FMT should only be used once current guidelines of vancomycin therapy have failed. Future randomized control trials of vancomycin taper versus FMT would be beneficial in determining which option is most effective for treating recurrence. Trials with larger sample sizes need to be performed to establish the effect that FMT has on the microbiome. Furthermore, longitudinal studies of pediatric patients would establish long-term efficacy of this treatment option. Given the small number of pediatric studies, establishing a national FMT database would be useful in analyzing the impact and side effects of FMT. Since this study was an observational study, we did not collect information on participants’ diet or environmental factors which may play a role in the gut microbiome. However, in future studies these determinants of health are important to consider in patients who undergo FMT and also in characterizing factors which affect the gut microbiome.

Conclusions

In our limited prospective study of children with rCDI, with and without IBD, FMT is efficacious, with minimal adverse effects, and improves the intestinal microbiome in favor of higher proportion of Bacteroidetes.

Supplemental Information

Table S1 Average relative abundance of significantly altered genera per FMT status

Wilcoxon signed rank test was performed for changes in genera average relative abundance values for all pair combination of FMT statuses. Any genus not found it at least 10% of the samples was removed from analysis. P-values were controlled for multiple hypothesis testing using Benjamini-Hochberg adjustment and adjusted p-value <0.05 was deemed to be significant. NS, Non-significant.

Click here for additional data file.

Data S1 Raw data for study participants

Worksheets contain the relational database of the study participants (de-identified of names and dates) along with the demographic, past medical history, and relevant antimicrobial therapy received.

Click here for additional data file.

We would like to thank: Dr. Sangita Ganesh (Georgia Institute of Technology) for her help with sample processing; Ms. Jonelle McKay, Dr. Robert C. Jerris, Dr. Mark Gonzales (Children’s Healthcare of Atlanta’s Clinical Microbiology Laboratory), for their assistance in the processing and proper storage of all biosamples used in the molecular analyses; Program managers Ms. Anaam Mohammed (Pediatric Emergency Medicine Associates, LLC) and Ms. Victoria Churchill (Morehouse School of Medicine) for co-ordinating the project across institutions; and finally, our heartfelt gratitude to the nurses and providers at Children’s Center for Digestive Health Care.

Additional Information and Declarations

Competing Interests

Author Contributions

Human Ethics

Data Availability

The authors declare there are no competing interests.

Shaaz Fareed performed the experiments, analyzed the data, prepared figures and/or tables, authored or reviewed drafts of the paper, approved the final draft.

Neha Sarode performed the experiments, analyzed the data, contributed reagents/materials/analysis tools, prepared figures and/or tables, authored or reviewed drafts of the paper, approved the final draft.

Frank J. Stewart conceived and designed the experiments, performed the experiments, analyzed the data, contributed reagents/materials/analysis tools, authored or reviewed drafts of the paper, approved the final draft.

Aneeq Malik and Elham Laghaie performed the experiments, analyzed the data, authored or reviewed drafts of the paper, approved the final draft.

Saadia Khizer authored or reviewed drafts of the paper, approved the final draft.

Fengxia Yan analyzed the data, prepared figures and/or tables, authored or reviewed drafts of the paper, approved the final draft.

Zoe Pratte performed the experiments, approved the final draft.

Jeffery Lewis conceived and designed the experiments, performed the experiments, analyzed the data, contributed reagents/materials/analysis tools, authored or reviewed drafts of the paper, approved the final draft, identified the patients with underlying GI condition; performed all the FMT.

Lilly Cheng Immergluck conceived and designed the experiments, performed the experiments, analyzed the data, contributed reagents/materials/analysis tools, prepared figures and/or tables, authored or reviewed drafts of the paper, approved the final draft.

The following information was supplied relating to ethical approvals (i.e., approving body and any reference numbers):

The following Institutional Review Boards provided approval for the conduct of this study:

Children’s Healthcare of Atlanta; Morehouse School of Medicine; Georgia Institute of Technology.

The following information was supplied regarding data availability:

The raw data, de-identified, has been included as a Supplemental file.

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
