# Peer review of "Applying fecal microbiota transplantation (FMT) to treat recurrent Clostridium difficile infections (rCDI) in children"

_PeerJ, doi:10.7717/peerj.4663_

## Round 0.1 · original submission · Minor Revisions

· Academic Editor

Minor Revisions

I have been able to acquire three independent reviews of your paper and they all agree the paper is well written and of high significance. One reviewer questions the sample sizes and would like to see >100 individuals. However, the other two reviewers did not see this as a concern. Nevertheless, there are a number of issues to take care of in a revision and they are detailed in the three reviews.

Reviewer 1 ·

Basic reporting

The paper is well written, with sufficient background information and citations to relevant literature. Most of the tables and figures are meaningful and self-explanatory.

Experimental design

The research question is well defined and sufficient details are provided for replication of the study.

Validity of the findings

The statistical analysis is appropriate in most cases. The authors realize and discuss the potential limitations of the work and have relevant future directions.

Additional comments

Comments:
Please clarify why the single patient received nasal fmt.
Please clarify how were the donors assigned to patients.
Add few citations for line 55-56
Curious as to why family members were not selected as donors like in citation 25.
Line 206, please state the actual p-value
Line 165, please mention version of greengenes database
Lines 168-170, at what taxonomic level was beta-diversity calculated?
Kindly do one of ANOSIM, ADONIS, MANOVA test in place of lines 198-201. Don’t really see the use of Figure 2.
Line 230, provide a table showing the genera that showed differential abundance.
Please discuss how your design and results compare against the studies (citations) 23-25.
Please perform multiple testing correction wherever appropriate.

·

Basic reporting

Human gut microbiome is an emerging area of research to investigate the role of microbiome in human healthy vs. diseased conditions. Clostridim difficile infections (CDI) are serious health issues leading to the deaths among adults and children. The infections are recurrent due to lack of efficient drug therapy. Fecal microbiota transplantation (FMT) is a novel therapy used to treat CDI but the therapy has been limited to the adults as far. This study shows, for the first time, the efficacy of FMT in pediatrics.

Experimental design

The specimen was collected from donors and study participants (age range from 21 months to 18 years) pre and 3 months post-treatment. The microbiome profiles of the donor and post-FMT patients were mostly similar whereas a significant difference was observed between microbiome profiles of pre and post FMT patient samples. The study, therefore, showed that the FMT significantly affected the gut microbiome compositions in all the patients. There was also the significant increase in Bacteriodes: Firmicutes ratio post-FMT which is the indicator of the healthy gut microbiome. The difference in the microflora was observed at taxon and genus level.

Validity of the findings

The article is very well-written. The authors have properly examined the study participants for the number of earlier CDI episodes, symptoms, ongoing antibiotic treatments and disease symptoms before FMT therapy. Post-FMT follow up was also done carefully for 15 study participants. The results were consistent with the earlier findings reported in adults. The significant taxonomic shifts of gut microflora were observed in the patients post-FMT. The high abundance of Proteobacteria in pre-FMT profiles vs. high abundance of Bacteroidetes in post-FMT profiles was also consistent with the previous studies. However, there are some limitations in the study as self-mentioned by the authors which should be highlighted:
1) The first limitation is the sample size which is very small to predict the efficiency and efficacy of the particular therapy and could be misleading. The study is properly validated and publishable only when the sample size is large (>100 samples).
2) The diet and environment play a major role in determining the gut microbiome as shown in previous studies. These are not taken into consideration in this study. I suggest including the diet and environmental factors also in the study.
3) A table showing the age, sex, diet, and present antibiotic status of the study participants should be there in supplementary data for making the study more informative for the readers.

Additional comments

In light of above comments, I suggest the authors to perform an extensive study on FMT with large sample size to prove this study highly relevant, impactful and publishable.

·

Basic reporting

This manuscript aims to answer questions regarding the efficacy of fecal microbiota transplantation (FMT) for recurrent C difficile infection (CDI) in children within the age range of 2-18 yrs. The authors demonstrate that administration of feces from healthy donors to the children with or without underlying IBD/Colitis reduces the incidence and recurrence of CDI and improves the livelihood mainly by increasing Bacteroidetes to Firmicutes, ratio post-FMT. The Methods, source of data and data abstraction, assessment, analysis are well described in the manuscript. Authors are commended for providing extensive raw data set, and thorough statistical analysis.

Experimental design

1) 1. The two important cohort study related to the FMT for rCDI could be included as they are pioneering studies
Russell et al., J Pediatr Gastroenterol Nutr. 2014 May;58(5):588-92. doi: 10.1097/MPG.0000000000000283.
Bakken et al, Clin Gastroenterol Hepatol. 2011 Dec;9(12):1044-9. doi: 10.1016/j.cgh.2011.08.014. Epub 2011 Aug 24.
2) Description of the donor age, gender in Table No.1 would be more informative.

3) Please include the specimen preparation for transplantation procedure in detail. Was the preparation done under ambient air or in the anaerobic atmosphere?

4) What was the stability of the FMT mediated engraftment of the microbial population?

Validity of the findings

The major limitation associated with the manuscript is the small sample size, It is unclear if findings are reproducible and credible due to real lack of association or small sample size.

Additional comments

1) What could be the predominant reasons behind increased incidences and recurrences of CDI in children in recent times?

2). What was the donor age range? Is it advantageous to use a young FMT donor as compared to the aged one? Also, what could be the impact of matching the gender and ethnicity of the donor to recipients or vice-versa?

3) As the microbial flora of young children are considerably different than the adults, does the FMT from the adults to treat children affects the length of healing/recovery as compared to FMT from adult to adult?

---

## Round 0.2 · accepted · Accept

· Academic Editor

Accept

Thank you for your careful revision. I sent it back to one of the previous reviewers and that reviewer and I are satisfied that you have done a great job accommodating the reviewers' concerns from the first round and that your study is now ready for acceptance. Congratulations.

# ·

Basic reporting

NA

Experimental design

NA

Validity of the findings

NA

Additional comments

NA